# Role of Arbuscular Mycorrhizal Fungi and Phosphate Solubilizing Bacteria in Improving Yield, Yield Components, and Nutrients Uptake of Barley under Salinity Soil

Abdurrahman S. Masrahi [1], Abdulrahman Alasmari [2], Mostafa G. Shahin [3], Alaa T. Qumsani [4], Hesham F. Oraby [5] and Mamdouh M. A. Awad-Allah [6,*]

1   Biology Department, Faculty of Science, Jazan University, Jazan 45142, Saudi Arabia
2   Department of Biology, Faculty of Science, University of Tabuk, Tabuk 47512, Saudi Arabia
3   Agronomy Department, Faculty of Agriculture, Ain Shams University, Hadayek Shoubra, Cairo 11241, Egypt
4   Biology Department, Al-Jumum University College, Umm Al-Qura University, Makkah 21955, Saudi Arabia
5   Deanship of Scientific Research, Umm Al-Qura University, Makkah 21955, Saudi Arabia
6   Rice Research Department, Field Crops Research Institute, Agricultural Research Center, Giza 12619, Egypt
*   Correspondence: momduhm@yahoo.com

**Abstract:** Barley (*Hordeum vulgare*, L.) is the fourth most important cereal crop in the world. Salinity decreases the productivity of plants grown under salinity conditions. It leads to deficiency and limited absorption of water and nutrients, ionic stress, oxidative stress, and osmotic imbalance. In saline soil, a field experiment was conducted to verify the effects of nine combinations among three levels of bio-fertilizers, i.e., control (without), arbuscular mycorrhizal fungi (AMF), and phosphate solubilizing bacteria (PSB), as well as three levels of phosphorus fertilizer recommended dose (RDP) on barley yield, its components and nutrients uptake, to evaluate the useful influences of these combinations to improve P management under salinity stress related to yield and its components as well as N, P, and K uptake in barley. Findings revealed that the combination AMF + 100% RDP improved plant height, length of spike, spikes weight, number of spikes plant$^{-1}$, weight of 1000-grain, straw yield, grain yield, uptake of nitrogen (N), phosphorus (P), potassium (K) in grain and uptake of nitrogen (N), phosphorus (P), potassium (K) in straw by 19.76, 33.21, 40.08, 33.76, 14.82, 24.95, 47.52, 104.54, 213.47, 168.24, 124.30, 183.59, and 160.84% in the first season, respectively. Meanwhile, the increase was 19.86, 29.73, 40.47, 39.94, 14.92, 24.95, 47.94, 104.73, 213.33, 168.64, 124.47, 183.86, and 161.09% in the second season, respectively. AMF showed greater efficiency and effectiveness compared to PSB in improving yield and its components for all studied traits. The results of principle component analysis indicated that all combinations except AMF + zero% RDP, PSB + zero% RDP, control + zero% RDP, and control + 66% RDP showed high scores on positive PC1, where all studied traits were high. Therefore, it is recommended to inoculate the soil with AMF or PSB with the addition of phosphate fertilizer at the recommended dose under salinity conditions, i.e., AMF + 100% RDP (T1) or AMF + 66% RDP (T2) or PSB + 100% RDP (T4). The use of bio-fertilizers has increased plant tolerance to salt stress, and this was evident from the increase in different traits with the use of treatments that include bio-fertilizers.

**Keywords:** barley; arbuscular mycorrhizal fungi; phosphate solubilizing bacteria; phosphorus fertilizer; salinity soil

## 1. Introduction

Barley (*Hordeum vulgare*, L.) is the fourth most important cereal crop in the world. The total cultivated area of barley in Egypt in the 2020 season reached approximately 29,295 hectares, and the total production exceeded 104,092 tons, with an average of 3.55 tons/ha [1]. Barley is characterized by its ability to grow under difficult environmental conditions such as drought and salinity compared with other cereal crops [2]. The productive capacity of cereal crops decreases under salinity conditions resulting in limited

water uptake, osmotic imbalances, ion stress, and oxidative stress [3–5]. In addition, salinity plays an important role in the excessive production of reactive oxygen species (ROS) such as singlet oxygen ($^1O_2$), superoxide ($O_2\bullet-$), hydroxyl free radical ($OH\bullet$), and hydrogen peroxide ($H_2O_2$) production within the plant [6]. Elevated ROS levels cause serious dysfunction in many processes, such as gene expression, hormonal homeostasis, signaling pathways, photosynthetic efficiency, and protein inactivation, inhibit the action of several enzymes involved in metabolic pathways, and reduce grain yield, as a result of lipid and DNA oxidation [7–10].

Barley can mitigate the harmful effect of salinity by excluding $Na^+$ from adsorption and accumulating $Na^+$ in tissues [3,11]. Although barley is a salinity-tolerant plant, some microorganisms should be added as adjuvants at the rhizosphere to enhance productivity and improve barley macronutrient absorption such as phosphorus.

Phosphorus (P) is one of the most three main fundamental nutrient elements necessary for plants. It is required as a constituent of nucleic acids, phospholipids, and adenosine triphosphate (ATP), involved in organizing several metabolic pathways in plants [12,13]. Through root transporters, plants can directly absorb phosphate from the soil, but low levels of available phosphate in the rhizosphere decrease the ability to absorb phosphate [14]. Phosphorus also helps plants cope with salinity stress, allowing them to utilize their carbohydrates for coding ions [15].

Salt stress reduces available phosphorus absorption, which leads to nutritional imbalance [16]. On the other hand, many researchers observed that adding phosphorus fertilizer enhanced the growth and yield of common beans [17] as well as sunflowers [18] under salinity stress. In this respect, it is beneficial and necessary to introduce salt-tolerant phosphate-solubilizing bacteria (PSB) as well as arbuscular mycorrhizal fungi (AMF), which can provide bioavailable P to plants by the mobilization of P that is bio-unavailable in soil. PSB improves plant nutrition absorption [19]. PSB (*Bacillus megaterium* var. phosphaticum) excrete acid phosphatases and phytases, which are beneficial for converting phosphate from an insoluble state into the soluble one that can be absorbed by roots [20]. One of the recognized mechanisms for phosphate solubilization is the production of organic acids by soil bacteria [19]. The PSB improved N, P, and K uptake and enriched the salt tolerance capacity in wheat [21] and maize [22].

Moreover, arbuscular mycorrhizal fungi (AMF) perform a symbiotic relationship between a fungus and the roots of a plant [13,23,24]. It is well-documented that mycorrhizal inoculation can increase the uptake of nutrients, growth of plants, improve yield quality, and enhance several abiotic stresses [13,25–29]. Mycorrhizae's beneficial impact on plant growth was attributed to its improved phosphorus uptake [30]. Additionally, AMF colonization can counteract the negative effects on $K^+$ and $Na^+$ absorption caused by salinity while reducing $Na^+$ translocation to shoot tissues [5,31]. Because of these benefits, A.M.F. is considered a good choice for bio-ameliorating saline soils.

The hypothesis was to use bio-fertilization to alleviate the harmful effects of salinity and to increase the absorption and availability of phosphorus while increasing productivity and yield. Thus, this investigation was conducted to evaluate the useful influences of the combinations between phosphorus fertilizer rates and arbuscular mycorrhizal fungi (AMF) as well as phosphate solubilizing bacteria (PSB) to improve P management under salinity stress related to yield and its components as well as N, P, and K uptake in barley.

## 2. Materials and Methods

### 2.1. Site Description

During the winter seasons of 2018/2019 and 2019/2020, two field experiments were carried out at Sakha Farm—Production Sector, Agricultural Research Center, Kafr El-Sheikh Governorate, Egypt (latitude 31°06′ N, longitude 30°56′ E) to study the effect of combinations between phosphorus fertilizer rates and arbuscular mycorrhizal fungi (AMF) as well as phosphate solubilizing bacteria (PSB) on yield, its components as well as nutrients uptake of barley (*Hordeum vulgare*, L. c.v. Giza-123) under saline soil. The soil of the study

site was saline-sodic clay soil, and its properties (0–30 cm from surface) are tabulated in Table 1, according to US Soil Taxonomy (Soil Survey Staff 1999) [32]. Average monthly climatic data for the location during barley growing seasons of 2018/2019 and 2019/2020 was illustrated in Table 2.

**Table 1.** The physical and chemical properties for the soil of study site.

| Season | Physical Property | | | | | | | Chemical Property | | | | | | |
| | | | | | | | | Soluble Cation (meq 100 g$^{-1}$ soil) | | | | Soluble Anions (meq 100 g$^{-1}$ soil) | | |
| | Sand% | Silt% | Clay% | pH | EC (dS m$^{-1}$) | SAR | ESP | Na$^+$ | K$^+$ | Ca$^{++}$ | Mg$^{++}$ | HCO$_3^-$ | Cl$^-$ | SO$_4^{--}$ |
|---|---|---|---|---|---|---|---|---|---|---|---|---|---|---|
| 2018/2019 | 28.34 | 23.45 | 48.21 | 8.21 | 10.53 | 18.64 | 42.23 | 43.40 | 1.14 | 9.86 | 29.63 | 58.30 | 40.90 | 14.30 |
| 2019/2020 | 25.32 | 26.44 | 48.24 | 8.22 | 10.65 | 18.76 | 42.21 | 43.70 | 1.15 | 9.88 | 29.65 | 58.60 | 40.30 | 14.60 |

**Table 2.** Means monthly of climatic parameters during barely growth and development at the study site (2018/2019 and 2019/2020 seasons).

| Month | Air Temperature (oc) | | | | Relative Humidity (%) | | Precipitation (mm) | |
| | Minimum | | Maximum | | | | | |
| | 2018/2019 | 2019/2020 | 2018/2019 | 2019/2020 | 2018/2019 | 2019/2020 | 2018/2019 | 2019/2020 |
|---|---|---|---|---|---|---|---|---|
| December | 12.7 | 11.2 | 25.9 | 22.7 | 33.3 | 30.1 | 1.08 | 0.62 |
| January | 11.4 | 10.0 | 24.5 | 19.8 | 45.4 | 41.7 | 2.07 | 2.24 |
| February | 10.1 | 9.3 | 22.7 | 21.2 | 43.5 | 40.5 | 5.35 | 5.78 |
| March | 12.9 | 11.2 | 24.3 | 23.2 | 42.9 | 43.7 | 0.65 | 0.51 |
| April | 13.7 | 15.5 | 25.2 | 26.1 | 50.8 | 50.6 | 0.00 | 0.00 |

### 2.2. Experimental Design and Management

The trial was set up as a randomized complete block design (RCBD) with nine treatments in three replicates. The treatments included AMF + 100% recommended dose of phosphorus fertilizer (T1), AMF + 66% recommended dose of phosphorus fertilizer (T2), AMF + zero% recommended dose of phosphorus fertilizer (T3), PSB + 100% recommended dose of phosphorus fertilizer (T4), PSB + 66% recommended dose of phosphorus fertilizer (T5), PSB + zero% recommended dose of phosphorus fertilizer (T6), control + 100% recommended dose of phosphorus fertilizer (T7), control + 66% recommended dose of phosphorus fertilizer (T8) and control + zero% recommended dose of phosphorus fertilizer (T9). The recommended dose of phosphorus fertilizer was 53.57 kg $P_2O_5$ ha$^{-1}$ as 357.14 Kg of calcium superphosphate (15.5% $P_2O_5$). The method described by Gerdemann and Nicholson [33] was used to extract mycorrhizal spores containing a mixture of the genera *Glomus* sp. and *Gigaspora* sp. from the rhizosphere and they were identified according to the key of Schenck and Perez [34]. Mycorrhizal spores and their carriers were a dry and water-insoluble powder and were distributed into the soil at 5 cm depth, immediately before sowing. The AMF inoculum application rate was 7 g m$^{-2}$ (3550 spores m$^{-2}$). The phosphate solubilizing bacteria (*Bacillus megaterium* var. phosphaticum), which commercially named (phosphorien) was used at the rate of 1.4 kg per 143 kg grains ha$^{-1}$. Inoculation with phosphorien was performed by coating barley grains using sticking substance (Arabic gum 5%) then sown and irrigated directly after inoculation. The area of the experimental unit was 15 m$^2$ (5 × 3 m), consisting of 15 rows each 0.20 m apart. Nitrogen fertilizer rate was 142.86 kg N ha$^{-1}$ added as ammonium nitrate (33.5% N) in three equal portions. The first portion was added 21 days after sowing (DAS), the second at 35 DAS, and the third at 50 DAS. Potassium fertilizer was added at the rate of 57.14 kg $K_2O$ ha$^{-1}$ as potassium sulphate (48% $K_2O$) in one dose with the 1st dose of nitrogen fertilizer. Barley grain was sown at a rate of 119 kg ha$^{-1}$ on the 1st of December for both seasons. The

preceding crop was rice in both seasons. All other cultural practices were followed as recommended for barley fields.

### 2.3. Recorded Data

At harvest after 120 days after sowing, ten plants from each plot were randomly selected to evaluate height of plants (cm), spike length (cm), spike weight (g), number of grains spike$^{-1}$, and weight of 1000-grain (g). In each experimental plot, all plants were harvested and then separated into straw and grains to determine the yield of straw and grain ha$^{-1}$. The samples of grains and straw were acquired from all experimental units and desiccated at 65 °C until stable weight and then pounded. The total N in straw and grains was estimated using the micro-Kjeldahl method according to AOAC [35]. Phosphorus content (P%) was estimated colorimetrically using chlorostannous reduced molybdophosphoric blue color method as described by Chapman and Parker [36]. The content of potassium (K%) was determined in the digested plant materials using the flame photometer according to page et al., [37]. The uptake of N, P, and K (kg ha$^{-1}$) was determined by the multiplication of grain or straw yield by their N%, P%, and K% content, respectively.

### 2.4. Statistical Analysis

The data were statistically analyzed using ANOVA of randomized complete block design as mentioned by Casella [38], using Costat software program Version 6.303 [39]. Duncan's multiple range test at 0.05 level of probability by Waller and Duncan [40] was used for comparing treatment means. The principal component analysis (PCA) was used to evaluate correlations among different variables. The number of components was designated as a function of eigenvalues (>1.0) and the variance explained (>80%). The PCA-Biplot analysis, consisting of two concepts, [41] was used for visual analysis. The PCA-Biplot graphic was created by JMP®, Version 13.2.0. SAS Institute Inc., Cary, NC, USA, 1989–2021 [42].

## 3. Results

### 3.1. Agronomic and Yield Attributed Traits

Barley yield and yield components were significantly affected by the combinations among three different percentages of the recommended dose of phosphorus fertilizer (RDP) and AMF or PSB or control (without adding) in both the 2018/19 and 2019/20 seasons. The combinations of AMF + phosphorus fertilizer (T1 and T2) gave the highest values for all studied traits, followed by the combinations of PSB + phosphorus fertilizer (Tables 3–5 and Figure 1). As presented in Table 3, significant changes in plant height, spike length, and spikes weight of barley were obtained among treatments in both seasons. In this respect, the tallest plants were obtained with AMF + 100% RDP (T1) and AMF + 66% RDP (T2), followed by PSB + 100% RDP (T4), and the increase was by 19.76, 17.76, and 12.96% in the first year, while the increase was by 19.86, 17.84, and 13.01% in the second year, respectively. Meanwhile, treatment control + zero% RDP (T9) produced the shortest plants in both seasons. The maximum increase in spike length at harvest was observed with the application of AMF + 100% RDP (T1) by 33.21 and 29.73% compared with control + zero% RDP (T9), which gave the lowest value of spike length in the first and second seasons, respectively. AMF + 100% RDP (T1) achieved the heaviest weight of spikes and significantly equaled with AMF + 66% RDP (T2), AMF + zero% RDP (T3), PSB + 100% RDP (T4), PSB + 66% RDP (T5), and control + 100% RDP (T7) while control + zero% RDP (T9) recorded the minimum spike weight in both seasons. The increase in the weight of the spike was by the percentage of 40.08, 37.65, 34.41, 35.22, 26.32, and 28.34% in the first year, while the percentage of increase was by 40.47, 37.74, 34.24, 35.41, 26.85, and 28.40% in the second year, respectively (Table 3).

**Table 3.** Plant height, spike length, and spikes weight of Giza 123 barley cultivar were influenced by the treatment with arbuscular mycorrhizal fungi (AMF), phosphate-solubilizing bacteria (PSB), and different percentages of the recommended dose of phosphorus fertilizers (RDP) in the 2018/2019 and 2019/2020 seasons.

| Treatments | Plant Height (cm) | | Spike Length (cm) | | Spikes Weight (g) | |
|---|---|---|---|---|---|---|
| | 2018/2019 | 2019/2020 | 2018/2019 | 2019/2020 | 2018/2019 | 2019/2020 |
| (T1) AMF + 100% RDP | 111.50 ± 1.32 a | 116.14 ± 1.34 a | 10.47 ± 0.12 a | 10.91 ± 0.13 a | 3.46 ± 0.16 a | 3.61 ± 0.17 a |
| (T2) AMF + 66% RDP | 109.63 ± 1.27 a | 114.19 ± 1.33 a | 10.13 ± 0.62 a | 10.56 ± 0.64 a | 3.40 ± 0.16 ab | 3.54 ± 0.17 ab |
| (T3) AMF + Zero% RDP | 100.65 ± 1.24 cd | 104.82 ± 1.32 cd | 9.70 ± 0.27 abc | 10.10 ± 0.29 abc | 3.32 ± 0.16 ab | 3.45 ± 0.16 ab |
| (T4) PSB + 100% RDP | 105.17 ± 1.09 b | 109.51 ± 1.15 b | 10.04 ± 0.21 ab | 10.46 ± 0.22 ab | 3.34 ± 0.16 ab | 3.48 ± 0.17 ab |
| (T5) PSB + 66% RDP | 103.30 ± 1.21 bc | 107.56 ± 1.26 bc | 9.68 ± 0.49 abc | 10.08 ± 0.51 abc | 3.12 ± 0.15 ab | 3.26 ± 0.16 ab |
| (T6) PSB + Zero% RDP | 101.07 ± 1.07 cd | 105.22 ± 1.14 cd | 8.94 ± 0.39 bcd | 9.30 ± 0.41 bcd | 2.58 ± 0.12 cd | 2.69 ± 0.13 cd |
| (T7) Control + 100% RDP | 101.40 ± 1.59 bcd | 105.56 ± 1.66 bcd | 9.38 ± 0.42 abc | 9.77 ± 0.43 abc | 3.17 ± 0.15 ab | 3.30 ± 0.16 ab |
| (T8) Control + 66% RDP | 99.23 ± 1.79 d | 103.30 ± 1.87 d | 8.85 ± 0.42 cd | 9.21 ± 0.44 cd | 2.96 ± 0.14 bc | 3.08 ± 0.15 bc |
| (T9) Control + Zero% RDP | 93.10 ± 0.75 e | 96.90 ± 0.80 e | 7.86 ± 0.26 d | 8.41 ± 0.27 d | 2.47 ± 0.12 d | 2.57 ± 0.12 d |

The values are the mean values ± standard error values. Different letters next to the values indicate significant differences at $p \leq 0.05$ according to Duncan's multiple range test.

**Table 4.** Number of spikes per plant and 1000-grain weight of Giza 123 barley cultivar were affected by treatment with arbuscular mycorrhizal fungi (AMF), phosphate-solubilizing bacteria (PSB), and different percentages of the recommended dose of phosphorus fertilizers (RDP) in the 2018/2019 and 2019/2020 seasons.

| Treatments | Number of Spikes Plant$^{-1}$ | | 1000-Grain Weight (g) | |
|---|---|---|---|---|
| | 2018/2019 | 2019/2020 | 2018/2019 | 2019/2020 |
| (T1) AMF + 100% RDP | 59.03 ± 2.80 a | 61.49 ± 2.93 a | 53.55 ± 1.10 a | 55.78 ± 1.16 a |
| (T2) AMF + 66% RDP | 57.61 ± 2.74 ab | 60.00 ± 2.86 ab | 52.64 ± 0.97 abc | 54.82 ± 1.01 abc |
| (T3) AMF + Zero% RDP | 55.53 ± 2.64 ab | 57.84 ± 2.75 ab | 52.14 ± 1.66 abc | 54.30 ± 1.73 abc |
| (T4) PSB + 100% RDP | 58.29 ± 2.77 ab | 60.70 ± 2.89 ab | 53.18 ± 1.00 ab | 55.38 ± 1.05 ab |
| (T5) PSB + 66% RDP | 56.17 ± 2.67 ab | 58.49 ± 2.78 ab | 52.77 ± 0.54 abc | 54.94 ± 0.57 abc |
| (T6) PSB + Zero% RDP | 47.44 ± 2.25 cd | 49.39 ± 2.35 cd | 50.06 ± 0.63 c | 52.12 ± 0.65 c |
| (T7) Control + 100% RDP | 53.60 ± 2.55 abc | 55.80 ± 2.65 abc | 53.05 ± 1.24 ab | 55.22 ± 1.27 ab |
| (T8) Control + 66% RDP | 51.02 ± 2.42 bcd | 53.11 ± 2.53 bcd | 50.40 ± 0.71 abc | 52.46 ± 0.72 abc |
| (T9) Control + Zero% RDP | 44.13 ± 2.10 d | 43.94 ± 2.19 d | 46.64 ± 1.02 d | 48.54 ± 1.06 d |

The values are the mean values ± standard error values. Different letters next to the values indicate significant differences at $p \leq 0.05$ according to Duncan's multiple range test.

**Table 5.** Biological yield and harvest index of barley Giza 123 cultivar were influenced by the treatment of arbuscular mycorrhizal fungi (AMF), phosphate-solubilizing bacteria (PSB), and different percentages of the recommended dose of phosphorus fertilizers (RDP) in the 2018/2019 and 2019/2020 seasons.

| Treatments | Biological Yield (Ton ha$^{-1}$) | | Harvest Index (%) | |
|---|---|---|---|---|
| | 2018/2019 | 2019/2020 | 2018/2019 | 2019/2020 |
| (T1) AMF + 100% RDP | 11.22 ± 0.29 a | 11.69 ± 0.31 a | 39.79 ± 1.88 abc | 39.82 ± 1.89 abc |
| (T2) AMF + 66% RDP | 10.20 ± 0.25 abc | 10.63 ± 0.26 abc | 43.31 ± 1.57 a | 43.32 ± 1.52 a |
| (T3) AMF + Zero% RDP | 9.19 ± 0.43 cd | 9.57 ± 0.46 cd | 40.84 ± 0.01 ab | 40.88 ± 0.03 ab |
| (T4) PSB + 100% RDP | 10.58 ± 0.5 ab | 11.02 ± 0.51 ab | 39.66 ± 0.02 abc | 39.69 ± 0.08 abc |
| (T5) PSB + 66% RDP | 9.27 ± 0.44 cd | 9.65 ± 0.46 cd | 43.16 ± 3.93 a | 43.12 ± 3.94 a |
| (T6) PSB + Zero% RDP | 9.53 ± 0.45 bcd | 9.92 ± 0.47 bcd | 39.16 ± 0.02 abc | 39.20 ± 0.02 abc |
| (T7) Control + 100% RDP | 11.02 ± 0.53 a | 11.47 ± 0.55 a | 37.05 ± 0.01 bc | 37.03 ± 0.06 bc |
| (T8) Control + 66% RDP | 9.65 ± 0.46 bcd | 10.04 ± 0.48 bcd | 37.87 ± 0.88 bc | 37.94 ± 0.91 bc |
| (T9) Control + Zero% RDP | 8.58 ± 0.40 d | 8.92 ± 0.42 d | 35.37 ± 0.06 c | 35.32 ± 0.04 c |

The values are the mean values ± standard error values. Different letters next to the values indicate significant differences at $p \leq 0.05$ according to Duncan's multiple range test.

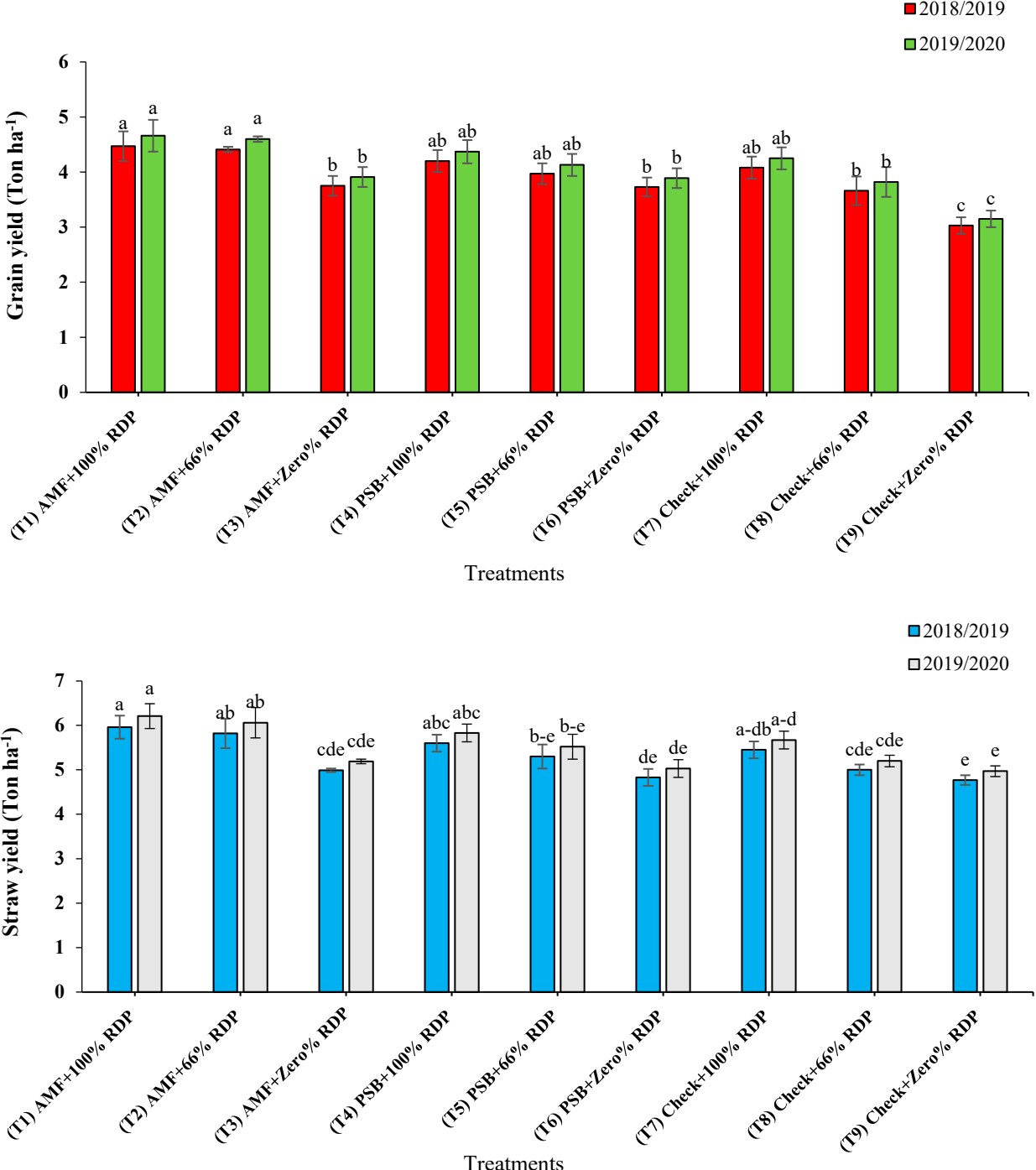

**Figure 1.** Grain and straw yield (Ton ha$^{-1}$) of barley Giza 123 cultivar were influenced by the treatment of arbuscular mycorrhizal fungi (AMF), phosphate-solubilizing bacteria (PSB), and different percentages of the recommended dose of phosphorus fertilizers (RDP) in the 2018/2019 and 2019/2020 seasons. Different letters indicate significant differences at $p \leq 0.05$ according to Duncan's multiple range test.

Concerning yield components (Table 4), no significant differences for the number of spikes plant$^{-1}$ and 1000-grain weight were realized among treatments except PSB + Zero% RDP (T6) and control + zero% RDP (T9). It is clear that those traits varied depending on adding phosphorus fertilizer. AMF has a higher efficiency than PSB under zero% RDP. In this connection, control + zero% RDP (T9) reduced the number of spikes plant$^{-1}$ by 25.24 and 28.54% compared with AMF + 100% RDP (T1), which recorded the highest

value in 2018/19 and 2019/20, respectively. The 1000-grain weight followed the same trend. Hence, the lightest weight of 1000-grain under control + zero% RDP (T9) was produced; meanwhile, the treatment AMF + 100% RDP (T1) produced the heaviest weight of 1000-grain in both seasons, followed by PSB + 100% RDP (T4). The percentage increase was 14.82 and 14.02% in the first year, and 14.92 and 14.09% in the second year, respectively (Table 4).

The combinations of AMF + phosphorus fertilizer (T1 and T2) gave the highest values for grain yield, followed by the combinations of PSB + phosphorus fertilizer (Figure 1). Whereas T1 significantly increased grain yield over control by 47.52 and 47.94% in the two seasons, respectively. In this regard, the increase compared with the recommended dose was 9.56 and 9.65%, respectively. Moreover, T2 significantly increased grain yield by 45.54 and 46.03% over control at the two studied seasons, respectively. This increase was 8.09 and 8.24% compared with the recommended dose, respectively. Moreover, T4 significantly augmented grain yield by 38.61 and 38.73% more than the control at the two seasons, respectively. Compared with the recommended dose, this increase was 2.94 and 2.82%, respectively. Furthermore, T5 significantly elevated grain yield by 31.02 and 31.11% in the two consecutive seasons, respectively. However, the decrease compared with the recommended dose was -2.70 and -2.82% in the first and second seasons, respectively (Figure 1). Complementary to these results, data presented in Figure 1 revealed that T1 significantly increased the straw yield by 24.95 and 24.95%, more than the control in both seasons, respectively. Meanwhile, the increase was 9.36 and 9.52% compared with the recommended dose. Moreover, T2 considerably enhanced the straw yield above the control by 22.01 and 21.93%, respectively. However, the percentages that increased over the recommended dose were 6.79 and 6.88%, respectively. When compared to control plants, T4 considerably elevated the straw yield by 17.40 and 17.30%, respectively. There was a 2.75 and 2.82% rise over the recommended dose, respectively. In the comparison with the control, T5 considerably boosted the straw yield by 11.11 and 11.07%, respectively. However, the difference fell below the recommended dose by −2.75 and −2.65% in the two seasons, respectively. Data depicted in Figure 1 indicated that the grain and straw yield were statistically influenced by the combinations of AMF, PSB, and phosphorus fertilizer rates. Application of AMF + 100% RDP (T1) increased grain yield by 50 and 46.88% compared with the lowest grain yield, which was produced by control + zero% RDP (T9) in the first and second seasons, respectively. Meanwhile, the treatment AMF + 100% RDP (T1) recorded the maximum straw yield and significantly leveled AMF + 66% RDP (T2), PSB + 100% RDP (T4), and control + 100% RDP in both seasons (Figure 1).

Available data in Table 5 showed biological yield and harvest index affected by the combination of AMF, PSB, and phosphorus fertilizer rates on barley. In both seasons, treatment AMF + 100% RDP (T1) achieved the highest biological yield with an increased percentage of 30.77 and 31.05% in the first and second seasons, respectively. Meanwhile, control + zero% RDP (T9) recorded the lowest. The difference between AMF + 100% RDP (T1), AMF + 66% RDP (T2), PSB + 100% RDP (T4), and control + 100% RDP (T7) did not reach the 0.05 level of significance in this respect. However, these treatments gave higher percentage values than the control) 30.77, 18.88, 23.31, and 28.44 in the first year, and 31.05, 19.17, 23.54, and 28.59 in the second year, respectively). Concerning harvest index, AMF + 66% RDP (T2) was the superior treatment for recording the higher value of harvest index but statistically leveled with AMF + 100% RDP (T1), AMF + zero% RDP (T3), PSB + 100% RDP (T4), PSB + 66% RDP (T5) and PSB + zero% RDP (T6) in the first and second seasons. This is an increase compared to the control by a percentage of 22.45, 12.50, 15.47, 12.13, 22.02, and 10.72% in the first year, while the percentage increase was 22.65, 12.74, 15.74, 12.37, 22.08, and 10.99% in the second year, respectively (Table 5).

### 3.2. Nutrients Uptake in Grain and Straw Yields

Statistically significant differences in grain nitrogen, phosphorus, and potassium uptake were obtained by the combinations between AMF, PSB, and phosphorus fertilizer rates

in the 2018/19 and 2019/20 seasons (Figure 2). In both seasons, treatments AMF + 100% RDP (T1) and AMF + 66% RDP (T2) recorded the highest grain N uptake, followed by PSB + 100% RDP (T4), while the treatment control + zero% RDP (T9) achieved the lowest values. The percentage increase compared to the control treatment (T9) was 104.54, 90.78, and 81.58% in the first year, while the increase was 104.73, 90.93, and 81.69% in the second year, respectively. Moreover, the treatments AMF + 100% RDP (T1) and AMF + 66% RDP (T2) recorded the highest grain P uptake in both seasons. With the increase over treatment (T9) by percentage was 213.47 and 192.63% in the first year, while the percentage was 213.33 and 192.53% in the second year, respectively. Moreover, grain K uptake recorded the highest values by AMF + 100% RDP (T1) and AMF + 66% RDP (T2), followed by all other combinations except control + zero% RDP (T9) in the first and second seasons. The percentage increase compared to the control treatment (T9) was 168.24, 150.26, 92.52, 104.27, 92.98, 81.17, 90.16, and 74.48% in the first year, while the increasing percentage was 168.64, 150.66, 92.81, 104.54, 93.19, 81.39, 90.35, and 74.57% in the second year, respectively.

There were significant variations between AMF, PSB, and phosphorus fertilizer rate combinations for straw N, P, and K uptake in the 2018/19 and 2019/20 seasons (Table 6). The treatment PSB + 100% RDP (T4) gave the highest value of N uptake in straw, followed by AMF + 100% RDP (T1), PSB + 66% RDP (T5), and control + 100% RDP (T7), while the treatment control + zero% RDP (T9) recorded the lowest value. The percentage increase compared to the control treatment (T9) was 151.84, 124.30, 138.40, and 131.69% in the first year, while the percentage was 151.98, 124.47, 138.47, and 131.75% in the second year, respectively. The highest values of uptake P in straw were recorded by AMF+100% RDP (T1), AMF + 66% RDP (T2), and PSB + 100% RDP (T4) in the first and second seasons. The percentage increase compared to the control treatment (T9) was 183.59, 177.54, and 151.95% in the first year, while the percentage increase was 183.86, 177.67, and 151.97% in the second year, respectively. Finally, AMF + 100% RDP (T1) achieved the maximum value of straw K uptake in both seasons and was statistically leveled with AMF + 66% RDP (T2) and PSB + 100% RDP (T4). The percentage increase compared to the control treatment (T9) was 160.84, 155.15, and 141.81% in the first year, while the increasing percentage was 161.09, 155.32, and 141.98% in the second year, respectively. Meanwhile, the treatment control + zero% RDP (T9) recorded the lowest value of K uptake in straw.

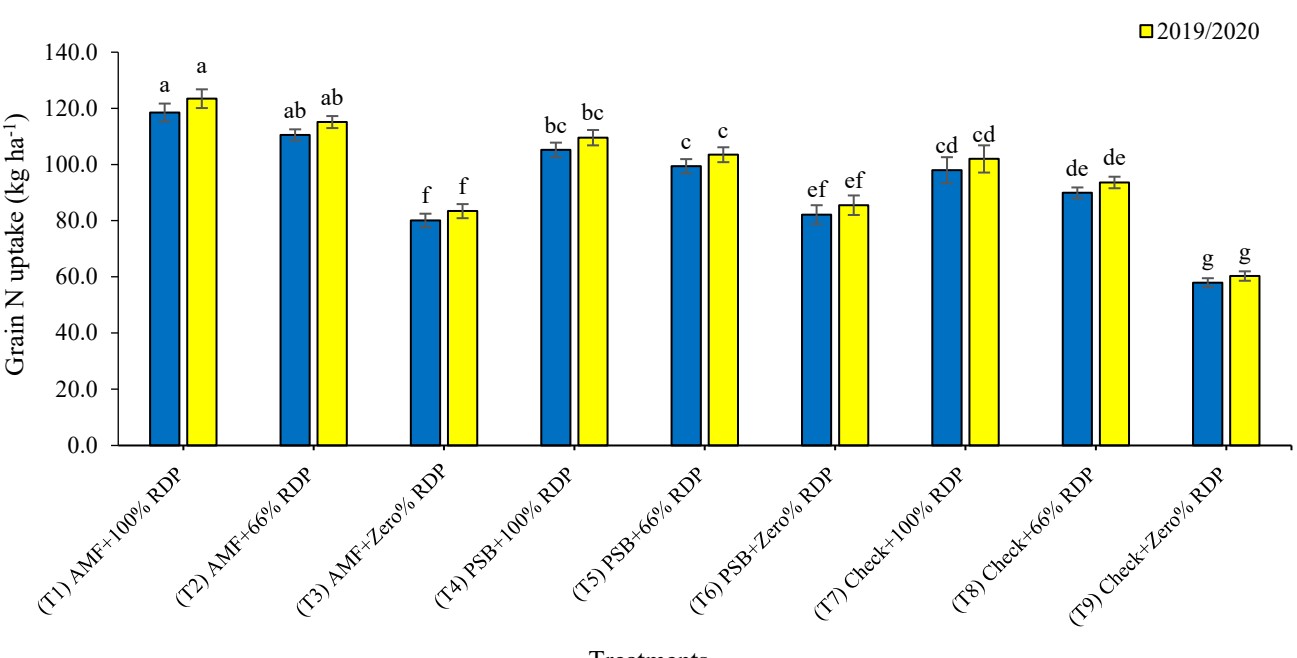

**Figure 2.** *Cont.*

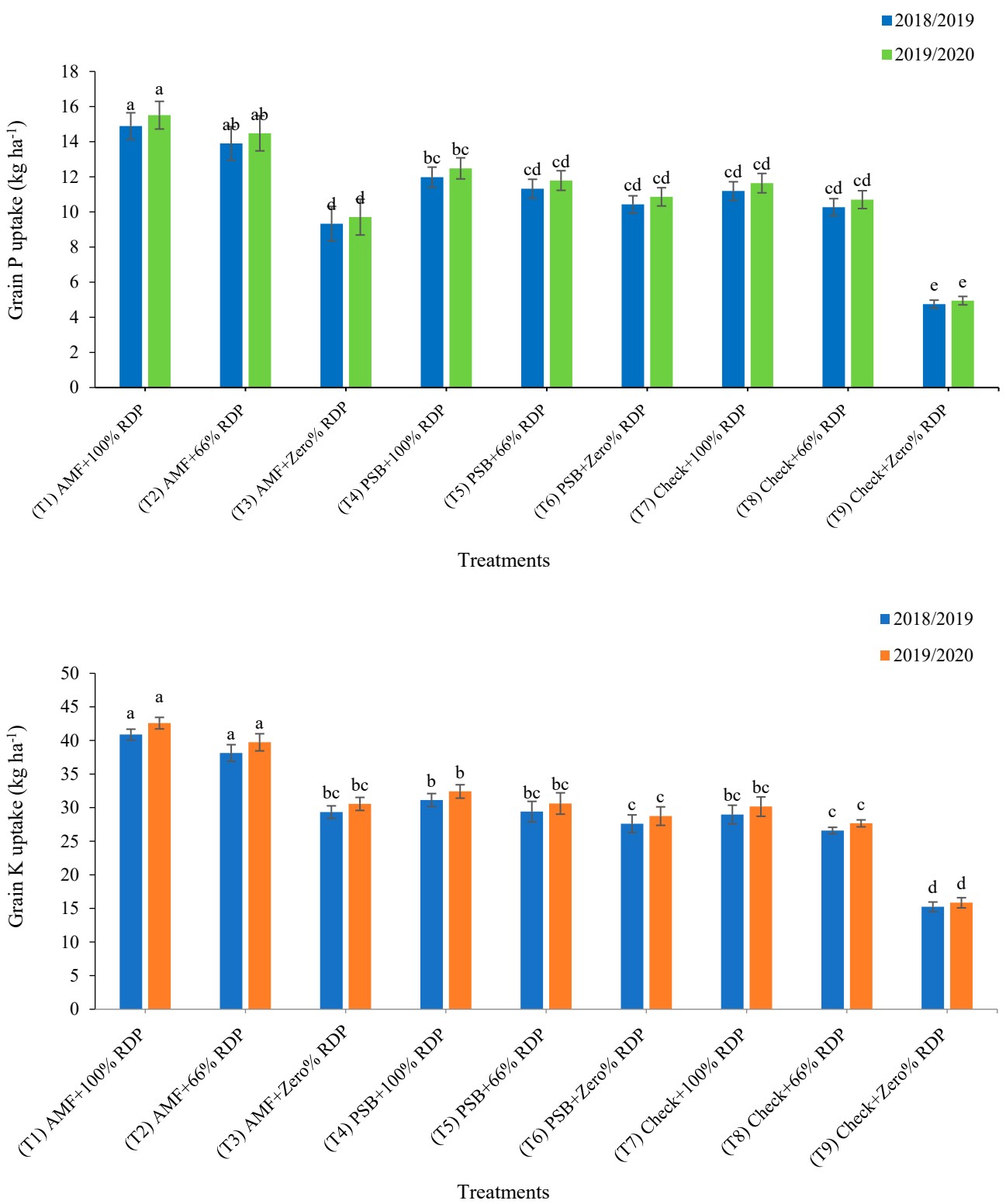

**Figure 2.** Grain N, P, and K uptake (kg ha$^{-1}$) of barley Giza 123 cultivar were influenced by the treatment of arbuscular mycorrhizal fungi (AMF), phosphate-solubilizing bacteria (PSB), and different percentages of the recommended dose of phosphorus fertilizers (RDP) in the 2018/2019 and 2019/2020 seasons. Different letters indicate significant differences at $p \leq 0.05$ according to Duncan's multiple range test.

**Table 6.** Nitrogen, phosphorus, and potassium uptake of Giza 123 barley cultivar as influenced by the treatment of arbuscular mycorrhizal fungi (AMF), phosphate-solubilizing bacteria (PSB), and different percentages of the recommended dose of phosphorus fertilizers (RDP) in the 2018/2019 and 2019/2020 seasons.

| Treatments | Straw N Uptake (kg ha$^{-1}$) | | Straw P Uptake (kg ha$^{-1}$) | | Straw K Uptake (kg ha$^{-1}$) | |
|---|---|---|---|---|---|---|
| | 2018/2019 | 2019/2020 | 2018/2019 | 2019/2020 | 2018/2019 | 2019/2020 |
| (T1) AMF + 100% RDP | 61.57 ± 2.95 ab | 64.13 ± 3.07 ab | 14.52 ± 0.77 a | 15.13 ± 0.8 a | 33.44 ± 1.38 a | 34.83 ± 1.44 a |
| (T2) AMF + 66% RDP | 60.21 ± 2.99 b | 62.71 ± 3.12 b | 14.21 ± 0.95 ab | 14.80 ± 0.99 ab | 32.71 ± 0.55 a | 34.06 ± 0.58 a |
| (T3) AMF + Zero% RDP | 33.79 ± 1.83 c | 35.18 ± 1.89 c | 10.48 ± 0.92 d | 10.92 ± 0.97 d | 24.77 ± 0.88 e | 25.79 ± 0.91 e |
| (T4) PSB + 100% RDP | 69.13 ± 3.28 a | 71.99 ± 3.42 a | 12.90 ± 0.98 abc | 13.43 ± 1.02 abc | 31.00 ± 0.97 ab | 32.28 ± 1.01 ab |
| (T5) PSB + 66% RDP | 65.44 ± 3.11 ab | 68.13 ± 3.24 ab | 12.21 ± 0.58 bcd | 12.71 ± 0.6 bcd | 29.34 ± 0.94 bc | 30.55 ± 0.97 bc |
| (T6) PSB + Zero% RDP | 59.61 ± 1.00 b | 62.06 ± 1.06 b | 10.48 ± 0.5 d | 10.91 ± 0.52 d | 22.93 ± 1.09 e | 23.88 ± 1.13 e |
| (T7) Control + 100% RDP | 63.60 ± 3.02 ab | 66.21 ± 3.15 ab | 11.79 ± 0.56 cd | 12.27 ± 0.58 cd | 28.02 ± 1.33 cd | 29.17 ± 1.39 cd |
| (T8) Control + 66% RDP | 58.35 ± 2.77 b | 60.74 ± 2.90 b | 10.83 ± 0.35 cd | 11.28 ± 0.36 cd | 25.71 ± 1.22 de | 26.76 ± 1.27 de |
| (T9) Control + Zero% RDP | 27.45 ± 1.30 c | 28.57 ± 1.36 c | 5.12 ± 0.24 e | 5.33 ± 0.25 e | 12.82 ± 0.61 f | 13.34 ± 0.63 f |

The values are the mean values ± standard error values. Different letters next to the values indicate significant differences at $p \leq 0.05$ according to Duncan's multiple range test.

### 3.3. Principal Component Analysis (PCA)

The PCA was used in order to evaluate the effect of the combinations between AMF, PSB, and phosphorous fertilizers rates affecting growth, yield, and its components, as well as N, P, and K uptake on barley plants (Figure 3). PCA results based on the correlation matrix were shown in biplots. Each variable is represented by an arrow, and the longer it is, the greater its contribution to a particular component. The angle between the arrows showed the correlation degree between the variables, where the smaller the angle, the greater the correlation, and vice versa. In this connection, it was found that the first and two principal components accounted for 84.00 and 7.24% of the variations for PC1 and PC2, respectively. The cumulative variance approached 89.24% of the total variance, meaning that this figure can describe nearly 90% of the data. Consequently, all studied traits contributed significantly to the variance for PC1 and PC2 as a result of increasing the lengths of arrows for all traits except for spikes weight plant$^{-1}$, 1000-grain weight, straw yield, and N uptake in straw. The results indicated that the combinations between control (without) or AMF or PSB as a bio-fertilizer and 100% of the recommended dose of phosphate fertilizer (RDP) showed high scores on positive PC1, where grain yield, straw yield, biological yield, N uptake in grain and straw, P uptake in grain and straw and K uptake in straw were high. Moreover, adding AMF or PSB as bio-fertilizer when plants were supplied with 66% RDP showed high scores on positive PC1, which was influential with respect to plant height, spike length, spikes weight plant$^{-1}$, number of spikes plant$^{-1}$, 1000-grains weight, K uptake in grains and harvest index. In contrast, treatments control + zero% RDP, AMF + zero% RDP and PSB + zero% RDP, as well as the combination among 66% RDP and without adding bio-fertilizer, exhibited high scores in negative PC1.

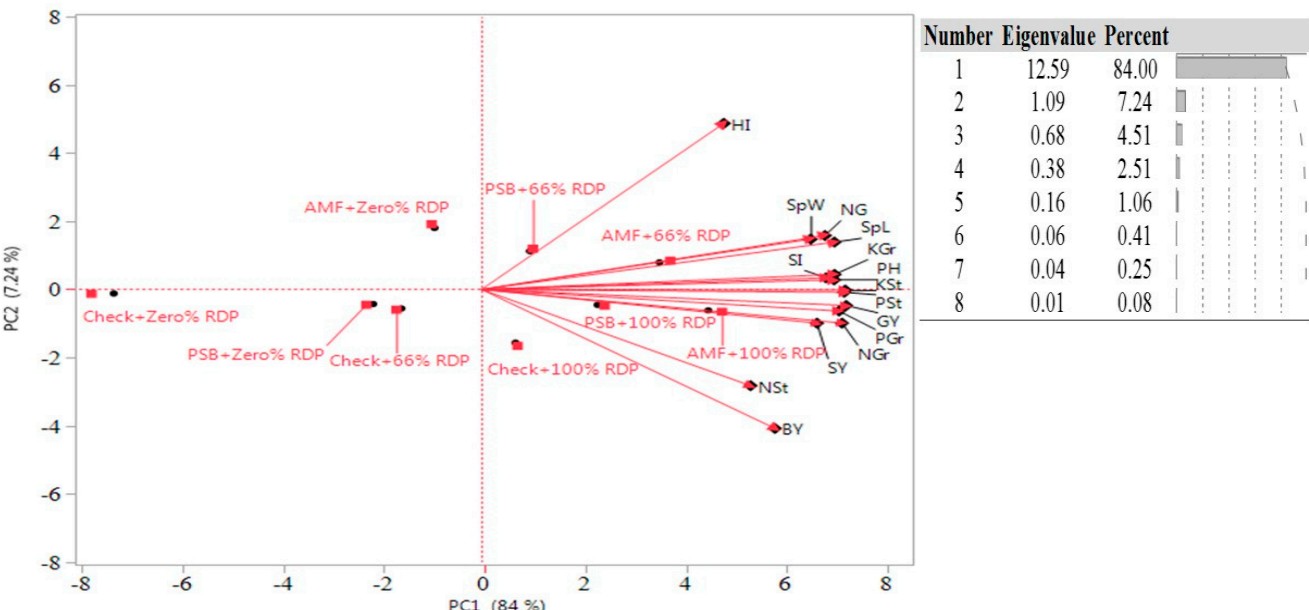

**Figure 3.** Biplot of principal component analysis based on the correlation matrix applied to plant growth traits, yield, and its components as well as biochemical traits. Abbreviations: PH, plant height; SpL, spike length; SpW, spikes weight; NG, number of spikes plant$^{-1}$; SI, 1000-grain weight; GY, grain yield; SY, straw yield; BY, biological yield; HI, harvest index; NGr, N uptake in grains; NSt, N uptake in straw; PGr, P uptake in grains; PSt, P uptake in straw; KGr, K uptake in grains and KSt, K uptake in straw. Different letters indicate significant differences at $p \leq 0.05$ according to Duncan's multiple range test.

## 4. Discussion

### 4.1. Agronomic and Yield Attributed Traits

The studied traits showed the highest values by applying the treatments T1 and T2, followed by T4 and T5 (Tables 3 and 4 and Figure 1). The inoculation by AMF or PSB significantly enhanced agronomic and yield-attributed traits than the control (without bio-fertilizer) treatment. In this respect, when combined with phosphorus, PSB has a significant function in cell division, root elongation, and is an essential component of ATP and ADP, which are responsible for higher yields. Furthermore, AMF inoculation recorded the highest results over PSB. The positive effect of AMF or PSB on agronomic and yield-attributed traits owing to reducing soil pH by organic acids realization and mineralized organic phosphorus. Moreover, organic materials in the soil can decompose more quickly due to fungus hyphae. Additionally, by enhancing the sink effect and moving photon-assimilates from the aerial sections to the roots, mycorrhizal fungi may have an impact on how much atmospheric $CO_2$ fixation occurs by host plants. Besides, the ability of AMF and PSB of producing or secreting some phytohormones led to enhanced agronomic, yield-attributed traits and grain yield [43–49].

Using a combination of AMF or PSB and phosphorus fertilizer rates increased yield and its components. The promotion effect of AMF and PSB as bio-fertilizers may be due to the effect of non-symbiotic phosphate, solubilizing microorganisms in exerting a positive influence on plant growth by the synthesis of phytohormones and enzymes (such as ACC deaminase) that modulate the plant hormonal level. Additionally, inorganic phosphate solubilization and organic phosphate mineralization phosphorus availability to plants [46–50].

Najafi et al. [49] revealed the beneficial impacts of microbial symbiosis on barley root growth as well as water and nutrient absorption. AMF promotes the growth of hair roots. As a result, the mycelium of the fungi grows more longitudinally and penetrates deeper layers of soil, increasing the availability of nutrients to plants [51,52]. It is implied that under

the conditions of salinity stress, endophytic fungi accompanying the plant can increase the sugar content as an osmotic protective substance [53,54]. The presence of elevated levels of polysaccharides in fungal plants indicates that they have a part in tolerance to salinity [54,55]. Several studies and previous reports have indicated that fungal endophytes guard plants amongst environmental stress by boosting their antioxidant activity. A supporter of this adjustment confirms the presence and accumulation of sugar, which may lead to the prevention of oxidative damage to cells or the elimination of ROS [54,56]. Furthermore, one of the most vital features responsible for the salinity tolerance potential is the increased accumulation of compatible organic solutes such as proline. Several previous studies have demonstrated that the accumulation of proline and soluble sugars in plants is mediated in various ways, including reducing the activity of proline E oxides, stimulating synthesis from feedstocks, and reducing protein structure partnership and protein destruction [57]. The remark of an escalation in proline due to AMF is consistent with the conclusions of [58] for Triticale. The AMF fungi significantly increased the photosynthesis of the host plants, and this consequently led to an increase in sugar content [59,60]. AMF treatment and inoculation increased chlorophyll content under normal cultivation conditions as well as under the salt-stressed ones [61]. Thus, changes in primary photosynthetic processes under conditions of environmental stress were evaluated using chlorophyll fluorescence as a robust and reliable noninvasive method [60,62,63].

High salt levels have a significant role in accelerating the oxidative damage to cell structures and membrane components, which could explain the greater value of EC at the highest salinity levels [64]. *Zea mays* inoculated with pseudomonas showed salt tolerance, which was facilitated by a reduction in electrolyte leakage, a rise in proline synthesis, and a conservation of leaf water content with $K^+$ ion-selective absorption [65]. Furthermore, iron-reduced reactive oxygen species damage and is considered crucial for maintaining membrane stability by improving antioxidant system activity in plants [66]. It has previously been documented that wheat plants exposed to various salt levels experience considerable reductions in soluble proteins [67]. Likewise, rice soluble protein contents in plants were significantly decreased as a result of various increased saline stress levels from 100, to 200, and 300 mmol/L of NaCl [68]. Previous studies suggested that salinity stress triggered a decline in protein synthesis [69–71], which is undoubtedly a key cause for the decrease in crude protein content. Different agricultural plants treated with endophytic fungus under either normal or salinity stress conditions showed high soluble protein contents [67,68]. Our findings in barley are consistent with those in different crops, and the activation of antioxidant enzymes is a great tactic for coping with salinity-induced oxidative stress [72]. The endogenous fungi play a major role in coping with and tolerating abiotic stress, specifically salinity stress tolerance [73,74]. Under salinity conditions, grain yield decreased in barley. The reason for this decrease in yield may have resulted from the reduction in the growth of stressed barley plants and the limitation of photosynthetic pigments in leaves [75–77]. The results of this study showed that inoculation with bio-fertilizer application under salinity stress led to a significant increase in barley grain yield. Other investigators suggested that co-inoculation AMF is an effective measure to increase plant growth and yield [78–80]. AMF creates fungal structures that aid in the interchange of inorganic minerals as well as carbon and phosphorus-containing molecules, giving host plants a significant boost in vigor. They can, therefore, greatly increase the phosphorus levels in both root and shoot systems. Mycorrhizal association enhances the phosphorus supply to the infected roots of host plants in phosphorus-limited environments. Improved growth frequency of AMF inoculation, which is directly associated with the intake of N, P, and carbon and moves towards roots and promotes the formation of tubers, is directly related to increased photosynthetic activities and other leaf functions. It has been noted that AMF maintains the uptake of P and N, thereby promoting plant development at greater and lower P levels under various irrigation regimes.

*4.2. Nutrients Uptake in Grain and Straw Yields*

The results presented in Table 6 and Figure 2 showed significant effects due to the bio-fertilizer uptake of nitrogen, phosphorus, and potassium in grain and straw of barley. Treatments containing AMF or PSB provided the highest values for all studied traits compared with control (without bio-fertilizer). In this respect, AMF recorded higher results over the PSB on the uptake of phosphorus and potassium in grains and straw yields. The positive effect of AMF or PSB on the uptake of phosphorus (P) and potassium (K) in grain and straw yields was due to the production of organic acids, which led to an increase in plant uptake of nutrients by reducing soil pH [79]. The data in Table 6 and Figure 2 showed that the maximum values of NPK uptake in grain and straw were obtained by the treatment of AMF + 100% RDP (T1) followed by AMF + 66% RDP (T2), PSB + 100% RDP (T4) and PSB + 66% RDP (T5), respectively. The application of AMF or PSB plays a significant role in the optimization of P solubilization, increasing nutrient levels, and mineralization of organic phosphate [14,80]. Moreover, the inoculation of AMF increases the buildup of dry matter and increases the uptake of water moisture, boosting plant tolerance to environmental challenges, including salinity and drought. Organic culturing for growth promotion and yield maximization can benefit considerably from the use of AMF for plant growth in different biological ecosystems. Nogueira et al. [79] explained the effect of AMF in increasing nutrient uptake in soybeans and also reported an important role of AMF in nitrogen uptake as a result of the indirect symbiosis between AMF and the plant. The treatment with AMF led to an increase in the uptake of nutrients such as P, Fe, and Mn by increasing the level of hyphae on the root surface, which, in turn, increased the uptake of nutrients independently of nitrogen uptake [79–84]. Calcium superphosphate has an effect on nutrient uptake, as the application of superphosphate resulted in improved NPK uptake because it contains some additional nutrients such as Ca, S, and other micronutrients in superphosphate. Moreover, calcium superphosphate provided phosphorous to the plant in the first stage of growth [85].

This investigation revealed the beneficial effects of combining phosphorus fertilizer rates and arbuscular mycorrhizal fungi (AMF) as well as phosphate solubilizing bacteria (PSB) to enhance P bioavailability under salinity stress. The applications improved yield and its components as well as N, P, and K uptake in barley.

## 5. Conclusions

Based on the presented results, it can be concluded that the application of bio-fertilizer has great potential to improve grain yield, barley production, and mineral content under salinity conditions. The study explained that the application of Mycorrhizal fungi or PSB plus superphosphate and indicated that the combined effect of AMF + 100% RDP and AMF + 66% RDP followed by PSB + 66% RDP resulted in the highest values of grain and straw yield without significant difference between the first and two treatments and low significant difference between the first two from second and third treatment. Thus, bio-fertilizers are considered a good tool for promoting barley growth, nutrient uptake, and yield of barley plants, particularly in saline soils. The combination of these two treatments indicated that the same yield can be obtained while saving a significant RDP amount. Therefore, it can successfully be exploited in conditions of the saline type of soils to improve barley production. The physiological characteristics and antioxidant enzymes, along with the quality traits, are under investigation for barley under stress for this work as well as for several other barley germplasm.

**Author Contributions:** Conceptualization, A.S.M., M.G.S., A.T.Q., H.F.O. and M.M.A.A.-A.; data curation, A.S.M., M.G.S., A.T.Q. and H.F.O.; formal analysis, A.S.M., A.A., M.G.S., A.T.Q. and M.M.A.A.-A.; funding acquisition, A.S.M. and H.F.O.; investigation, A.S.M., A.A., A.T.Q. and H.F.O.; methodology, A.A., M.G.S., A.T.Q. and M.M.A.A.-A.; project administration, H.F.O. and M.M.A.A.-A.; resources, M.M.A.A.-A.; software, A.S.M., A.A., M.G.S. and A.T.Q.; supervision, M.M.A.A.-A.; validation, A.A., M.G.S. and H.F.O.; visualization, A.A. and H.F.O.; writing—original draft, A.S.M.,

M.G.S., A.T.Q. and M.M.A.A.-A.; writing—review and editing, A.A., M.G.S. and M.M.A.A.-A., all authors; funding, H.F.O. All authors have read and agreed to the published version of the manuscript.

**Funding:** The author (Hesham Oraby) extends his appreciation to the Deputyship for Research & Innovation, Ministry of Education in Saudi Arabia for funding this research work through the project number: IFP22UQU4350043DSR53.

**Institutional Review Board Statement:** Not applicable.

**Informed Consent Statement:** Not applicable.

**Data Availability Statement:** The data used to support the findings of this study are included within the article.

**Conflicts of Interest:** The authors declare no conflict of interest.

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
