# Peer review of "Role of Arbuscular Mycorrhizal Fungi and Phosphate Solubilizing Bacteria in Improving Yield, Yield Components, and Nutrients Uptake of Barley under Salinity Soil"

_agriculture, doi:10.3390/agriculture13030537_

Round 1
Reviewer 1 Report
Dear appreciated Authors,
The manuscript “Role of Arbuscular Mycorrhizal Fungi and Phosphate Solubilizing Bacteria in Improving Yield, Yield components and Nutrients Uptake of Barley under Salinity Soil” was found interesting, appears to be scientifically sound and generally well-written. The topic provides important information about the mode of reduction the harmful effect of salinity soils and improving barley production in these conditions.
There are a certain details which should be considered and minor revisions have to be made before it can reach a publishable value. To improve the quality of the manuscript, I propose the following suggestions:
Line 21: Please use word "control" instead of "check"
Line 412: Based on the presented results it can be concluded that the application of bio-fertilizer has the great potential to improve grain yield, barley production and mineral content under salinity condition.
Line 418: Combination of these two treatments indicated that the same yield can be obtained while saving one third (33%) of the RDP amount and reserving a considerable part of the production cost. Therefore, it can be successfully exploited in conditions of saline type of soils to improve barley production.
Based on all, author's paper should be accepted for publication after minor revisions, because the paper represents a significant contribution for improvement managements to increase the barley production in conditions of saline type of soils.
Best regards,
NL
Author Response
Reviewer 1
Author's Reply to the Review Report (Reviewer 1)
Comments and Suggestions for Authors
Dear appreciated Authors,
The manuscript “Role of Arbuscular Mycorrhizal Fungi and Phosphate Solubilizing Bacteria in Improving Yield, Yield components and Nutrients Uptake of Barley under Salinity Soil” was found interesting, appears to be scientifically sound and generally well-written. The topic provides important information about the mode of reduction the harmful effect of salinity soils and improving barley production in these conditions.
There are a certain detail which should be considered and minor revisions have to be made before it can reach a publishable value. To improve the quality of the manuscript, I propose the following suggestions:
Line 21: Please use word "control" instead of "check"
Thank you very much. We have made the change as suggested by the reviewer.
Line 412: Based on the presented results it can be concluded that the application of bio-fertilizer has the great potential to improve grain yield, barley production and mineral content under salinity condition.
Thank you very much. We have made the change as suggested by the reviewer.
Line 418: Combination of these two treatments indicated that the same yield can be obtained while saving one third (33%) of the RDP amount and reserving a considerable part of the production cost. Therefore, it can be successfully exploited in conditions of saline type of soils to improve barley production.
Response: Thank you. Corrected as suggested by the reviewer
Based on all, author's paper should be accepted for publication after minor revisions, because the paper represents a significant contribution for improvement managements to increase the barley production in conditions of saline type of soils.
Best regards,

Reviewer 2 Report
Line 82-85 Given the results of many previous studies listed above, could you summarize the necessity of your meta-analysis study? Add before this paragraph.
Line 94 The properties were at what depth of soil?
Line 139-140 What was the sampling time?
Line 170 change “by” to “followed”
Table 4 “T2: 60.00…T3: 54.30…T4: 60.7… T7: 55.8…” Whether the zero at the end of the decimal point is retained, please be unified.
Figure 3 What are the red and black dots? Two seasons?
Line 306-328 There are too many statements about results in these two paragraphs
Line 339-341 “Additionally, the benefits of using bio-fertilizers and the desire to adopt novel techniques enhanced soil fertility and yield productivity.” This sentence does not explain how AMF can improve soil fertility and yield, right? Suggested to delete.
Line 413-417 “Thus, the bio-fertilizers are considered a good tool for promoting barley growth, nutrients uptake, and yield of barley plants particularly in saline soils.” What about PSD?
Line 418-420 There is no discussion of production cost above.
Moreover, the different letters next to all the values which indicate significant differences at p≤ 0.05 according to Duncan's multiple range test were the same in both 2018/19 and 2019/20 seasons in Table 3, Table 4, Table 5, Table 6, Figure 1 and Figure 2, is it a coincidence?
Author Response
Reviewer 2
Author's Reply to the Review Report (Reviewer 2)
Comments and Suggestions for Authors
Line 82-85 Given the results of many previous studies listed above, could you summarize the necessity of your meta-analysis study? Add before this paragraph.
Line 94 The properties were at what depth of soil? Soil samples were taken at a depth of 0 - 30 cm (30 cm from the surface).
Line 139-140 What was the sampling time? At harvest after 120 DAS (days after sowing)
Line 170 change “by” to “followed”
Thank you very much. We have made the change as suggested by the reviewer.
Table 4 “T2: 60.00…T3: 54.30…T4: 60.7… T7: 55.8…” Whether the zero at the end of the decimal point is retained, please be unified.
Thank you very much. We corrected as suggested by the reviewer.
Figure 3 What are the red and black dots? Two seasons?
Yes, they are for the two seasons.
Line 306-328 There are too many statements about results in these two paragraphs :
Response: Thank you very much. They were transferred to the results section as suggested by the reviewer.
Line 339-341 “Additionally, the benefits of using bio-fertilizers and the desire to adopt novel techniques enhanced soil fertility and yield productivity.” This sentence does not explain how AMF can improve soil fertility and yield, right? Suggested to delete.
Response: thank you. The sentence was deleted as suggested.
Line 413-417 “Thus, the bio-fertilizers are considered a good tool for promoting barley growth, nutrients uptake, and yield of barley plants particularly in saline soils.” What about PSD?
Response: Thank you. It was added as suggested by the reviewer.
Line 418-420 There is no discussion of production cost above.
Response: Thank you. It is corrected as suggested by the reviewer.
Moreover, the different letters next to all the values which indicate significant differences at p≤ 0.05 according to Duncan's multiple range test were the same in both 2018/19 and 2019/20 seasons in Table 3, Table 4, Table 5, Table 6, Figure 1 and Figure 2, is it a coincidence?
Response: Thank you. Statistical analysis was performed for each year separately.
Best regards,

Reviewer 3 Report
Dear editor,
The manuscript evaluates the effects of some bio fertilizers such as AMF and phosphate solubilizing bacteria (PSB) on the productivity of barley plant under salinity soil conditions. The manuscript idea is good. However, to improving the manuscript quality, some attributes should be added to the article. In the salinity or stressful conditions, the physiological parameters of plants, especially barley, was changed. It is clear that the yield of all plants will decrease under salt stress conditions. Under the conditions of salt stress, the physiological parameters of the plant will undergo important changes. I suggest that the physiological parameters and activity level of antioxidant enzymes be added to the article. The authors should answer the question whether the application of biofertilizers has been able to increase the plant's resistance to salt stress or not? Additionally, I can see too many paragraphs in discussion section that are reporting the previous studies reports. This is not a really discussion. Therefore, my final recommendation is "Major revision".
Abstract
- Please add one or two sentences as abstract background about the negative impacts of salinity conditions on the plants productivity and also experiment objectives.
- Underscore the scientific value-added to your paper in your abstract. Your abstract should clearly state the essence of the problem you are addressing, what you did and what you found and recommend. That will help a prospective reader of the abstract to decide if they wish to read the entire article.
- L25: The authors noted that ‘the combination AMF+100% RDP improved plant height, length of spike, spikes weight, number of …’. The sentence is not clear. The authors should be added how many percentages increased or decreased.
- what is the best recommendation for barley growers?
Introduction
- Please add the cultivation area and average productivity of barley in your country.
- The linkage between paragraphs is missed.
- The novelty should be highlighted in introduction and discussion section.
- The hypothesis should be added at the end of introduction section
Materials and methods
- L109-118: In the recommendation dose, the authors use only P fertilizers?
- L119: The citation format is not true.
- L121: The citation format is not true.
- Do you measure the AMF colonization rate? Also, do you have a microscopic picture from AMF hyphae on the root cells?
- Where was the phosphorien fertilizer prepare from?
- L141: The harvesting method was randomly?
Results
- The results section is good. However, the authors should be added the increasing or decreasing percentage of studied traits under different experimental treatments.
- Why the yield component and yield of barley in the second year was higher than first one?
- In the salinity or stressful conditions, the physiological parameters of plants, especially barley, was changed. It is clear that the yield of all plants will decrease under salt stress conditions. Under the conditions of salt stress, the physiological parameters of the plant will undergo important changes. I suggest that the physiological parameters and activity level of antioxidant enzymes be added to the article. The authors should answer the question whether the application of biofertilizers has been able to increase the plant's resistance to salt stress or not?
Discussion
- L306-328: This section is results of the study and should be added in results section. This is not a discussion. Please add the main reasons for increasing or decreasing the studied traits based on the obtained results and experimental treatments.
- I can see too many paragraphs in this section that are reporting the previous studies reports. This is not a really discussion. Please try your best to illustrate the possible reasons after this.
- L 338: The citation format is not true. Please check all text.
Conclusion
- This section is repetitive and should be rewritten.
- Please make sure your conclusions' section underscores the scientific value-added of your paper, and/or the applicability of your findings/results. Highlight the novelty of your study.
- What suggestions do you have for future research in this field?
Author Response
Reviewer 3
Comments and Suggestions for Authors
Dear editor,
The manuscript evaluates the effects of some bio fertilizers such as AMF and phosphate solubilizing bacteria (PSB) on the productivity of barley plant under salinity soil conditions. The manuscript idea is good. However, to improving the manuscript quality, some attributes should be added to the article. In the salinity or stressful conditions, the physiological parameters of plants, especially barley, was changed. It is clear that the yield of all plants will decrease under salt stress conditions. Under the conditions of salt stress, the physiological parameters of the plant will undergo important changes. I suggest that the physiological parameters and activity level of antioxidant enzymes be added to the article.
We would like to thank the reviewer for this valuable suggestion. In fact, the physiological characters and antioxidant enzymes along with the quality traits are under investigation right now for barley under stress for this work as well as for several other barley germplasm. This part was added as future research in the field.
The authors should answer the question whether the application of biofertilizers has been able to increase the plant's resistance to salt stress or not?
The use of biofertilizers increased plant tolerant to salt stress, and this was evident from the increase in different traits with the use of treatments that include biofertilizers. This part was added in the abstract and conclusion.
Additionally, I can see too many paragraphs in discussion section that are reporting the previous studies reports. This is not a really discussion. Therefore, my final recommendation is "Major revision".
Response: Thank you. Corrected as suggested by the reviewer.
Abstract
- Please add one or two sentences as abstract background about the negative impacts of salinity conditions on the plants productivity and also experiment objectives.
Response: Thank you. We added the sentences as suggested by the reviewer.
- Underscore the scientific value-added to your paper in your abstract. Your abstract should clearly state the essence of the problem you are addressing, what you did and what you found and recommend. That will help a prospective reader of the abstract to decide if they wish to read the entire article.
Response: Thank you. Some sentences were added as suggested by the reviewer.
- L25: The authors noted that ‘the combination AMF+100% RDP improved plant height, length of spike, spikes weight, number of …’. The sentence is not clear. The authors should be added how many percentages increased or decreased.
Response: Thank you. The percentages were added as suggested by the reviewer.
- what is the best recommendation for barley growers?
AMF+100% RDP (T1) or AMF+66% RDP (T2) or PSB+100% RDP (T4)
The recommended treatments were added to the conclusion.
Introduction
- Please add the cultivation area and average productivity of barley in your country.
The cultivated area and average productivity of barley in the country was added as suggested by the reviewer.
- The linkage between paragraphs is missed.
Response: Thank you. The link between paragraphs was added.
- The novelty should be highlighted in introduction and discussion section.
Response: Thank you. The hypothesis and work novelty were added at the end of the introduction and discussion section as suggested by the reviewer.
- The hypothesis should be added at the end of introduction section Response: Thank you. The hypothesis was added at the end of the introduction section.
Materials and methods
- L109-118: In the recommendation dose, the authors use only P fertilizers?
Response: Thank you very much. In this paragraph (L109-118), The recommended dose of phosphorus (P) was mentioned. Meanwhile, the recommended dose of nitrogen (N) and potassium (K) was mentioned in the paragraph from line No. 129-132
- L119: The citation format is not true.
Response: Thank you. Corrected as suggested by the reviewer.
- L121: The citation format is not true.
Response: Thank you. Corrected as suggested by the reviewer.
- Do you measure the AMF colonization rate?
Response: Thank you very much. Roots in treated experimental plots were examined for infection in the presence of AMF structures, such as arbuscules, vesicles, interstitial sutures, and additional root hyphae collected from each slide. We would like to assure that it will be taken into consideration in any related coming work.
Also, do you have a microscopic picture from AMF hyphae on the root cells? Thankyou very much for the suggestion. Unfortunately, a microscopic picture of AMF hyphae on root cells is not available at the moment. We would like to assure that it will also be taken into consideration in any related coming work.
- Where was the phosphorien fertilizer prepare from?
Phosphorine bio-fertilizer was prepare in Soils and Water Research Institute, Agriculture Research Center, Giza, Egypt, and produced by General Organization for Agric. Equalization Fund (G.O.A.E.F.)
- L141: The harvesting method was randomly?
The method of harvesting was not random. Ten plants were taken without selection or bias from each experimental plot to estimate the components of the yield and other traits that are estimated at and after harvest.
Results
- The results section is good. However, the authors should be added the increasing or decreasing percentage of studied traits under different experimental treatments.
Response: Thank you very much. Increasing or decreasing percentage were added as suggested by the reviewer.
- Why the yield component and yield of barley in the second year was higher than first one?
Response: Thank you. The increase could be due to reduction in air temperature and the slight increase in precipitation during the vegetative growth period of the plant in months January and February.
- In the salinity or stressful conditions, the physiological parameters of plants, especially barley, was changed. It is clear that the yield of all plants will decrease under salt stress conditions. Under the conditions of salt stress, the physiological parameters of the plant will undergo important changes. I suggest that the physiological parameters and activity level of antioxidant enzymes be added to the article. The authors should answer the question whether the application of biofertilizers has been able to increase the plant's resistance to salt stress or not?
We would like to thank the reviewer for this valuable suggestion. In fact, the physiological characters and antioxidant enzymes along with the quality traits are under investigation right now for barley under stress for this work as well as for several other barley germplasm. This part was added as future research in the field.
The use of biofertilizers indeed increased plant tolerant to salt stress, and this is evident from the increase in different traits with the use of treatments that include biofertilizers. This part was added in the abstract and conclusion.
Discussion
- L306-328: This section is results of the study and should be added in results section. This is not a discussion. Please add the main reasons for increasing or decreasing the studied traits based on the obtained results and experimental treatments.
- I can see too many paragraphs in this section that are reporting the previous studies reports. This is not a really discussion. Please try your best to illustrate the possible reasons after this.
Response: Thank you. Corrected as suggested by the reviewer.
- L 338: The citation format is not true. Please check all text.
Response: Thank you. Corrected as suggested by the reviewer.
Conclusion
- This section is repetitive and should be rewritten.
Response: Thank you. Corrected as suggested by the reviewer.
- Please make sure your conclusions' section underscores the scientific value-added of your paper, and/or the applicability of your findings/results. Highlight the novelty of your study.
Response: Thank you. Corrected as suggested by the reviewer.
- What suggestions do you have for future research in this field?
Response: Thank you. Future research has been added to the end of conclusion.
Best regards,

Reviewer 4 Report
The paper is good and can be accepted. Minor revisions were required as follow:
The font in Figure 2 and Figure 3 is not in high quality. Please make the figures better.
Author Response
Reviewer 4
Author's Reply to the Review Report (Reviewer 4)
The paper is good and can be accepted. Minor revisions were required as follow:
The font in Figure 2 and Figure 3 is not in high quality. Please make the figures better.
Thank you very much, we have made the change as suggested by the reviewer
Best regards,

Round 2
Reviewer 3 Report
Dear editor,
In the revised version, the authors have appropriately edited and revised this earlier version according to the comments and suggestions from the reviewers and have reasonably addressed most of the concerns and issues in the review reports. The current version could be accepted for publication in this journal.
Best regards